# Energy Efficiency Analysis of the Refining Unit in Thermo-Mechanical Pulp Mill

Behnam Talebjedi [1,*], Timo Laukkanen [1], Henrik Holmberg [1], Esa Vakkilainen [2] and Sanna Syri [1]

[1] Department of Mechanical Engineering, School of Engineering, Aalto University, 14400 Espoo, Finland; timo.laukkanen@aalto.fi (T.L.); henrik.holmberg@aalto.fi (H.H.); sanna.syri@aalto.fi (S.S.)
[2] Department of Energy, Lappeenranta University of Technology, 95992 Lappeenranta, Finland; esa.vakkilainen@lut.fi
[*] Correspondence: Behnam.talebjedi@aalto.fi

**Abstract:** A refining model is developed to analyses the refining process's energy efficiency based on the refining variables. A simulation model is obtained for longer-term refining energy analysis by further developing the MATLAB Thermo-Mechanical Pulping Simulink toolbox. This model is utilized to predict two essential variables for refining energy efficiency calculation: refining motor-load and generated steam. The conventional variable for presenting refining energy efficiency is refining specific energy consumption (RSEC), which is the ratio of the refining motor load to throughput and does not consider the share of recovered energy from the refining produced steam. In this study, a new variable, corrected refining specific energy consumption (CRSEC), is introduced and practiced for better representation of the refining energy efficiency. In the calculation process of the CRSEC, recovered energy from the refining generated steam is considered useful energy. The developed model results in 160% and 78.75% improvement in simulation model determination coefficient and error, respectively. Utilizing the developed model and hourly district heating demand for CRSEC calculation, results prove a 22% annual average difference between CRSEC and RSEC. Findings confirm that the wintertime refining energy efficiency is 27% higher due to higher recovered energy in the heat recovery unit compared to summertime.

**Keywords:** thermo-mechanical pulping; district heating; pulping energy efficiency; refining energy efficiency; data analysis

## 1. Introduction

Energy efficiency analysis and improvement are of the most important issues for energy-dependent industries for the transition toward a more sustainable and energy-efficient industry. Most often, their level of production and profitability are strongly affected by the reliability of energy supply as well as fluctuations in energy prices [1,2]. The pulp and paper industry is in the fourth place of the most extensive industrial energy consumer worldwide [3]. Almost 6% of total industrial energy consumption and 2% of total industrial carbon dioxide emission is attributed to the pulping and papermaking processes [3,4]. In 2007, 6.87 EJ of final energy was consumed in the pulp and paper industry [5–8]. Today, the most dominant mechanical pulping process is the thermo-mechanical pulping process (TMP) [9]. Figure 1 shows a typical flow diagram for a TMP plant [10]. This process is one of the most energy-consuming processes in the pulp and paper industry, with an energy efficiency of 10%–15% [11]. In terms of energy, unlike chemical pulping, which is heat-intensive, mechanical pulping is more electricity-intensive [5,12]. Due to the recent increase in the electricity cost in some countries, the TMP process profitability has been heavily decreased because of the massive electrical energy consumption. Moreover, environmental considerations and carbon emissions taxes have obligated pulp mills to reduce their energy consumption and carbon footprint [11–13]. Furthermore, more improved and uniform pulp quality is required for pulp mills to keep their position in the market share [14].

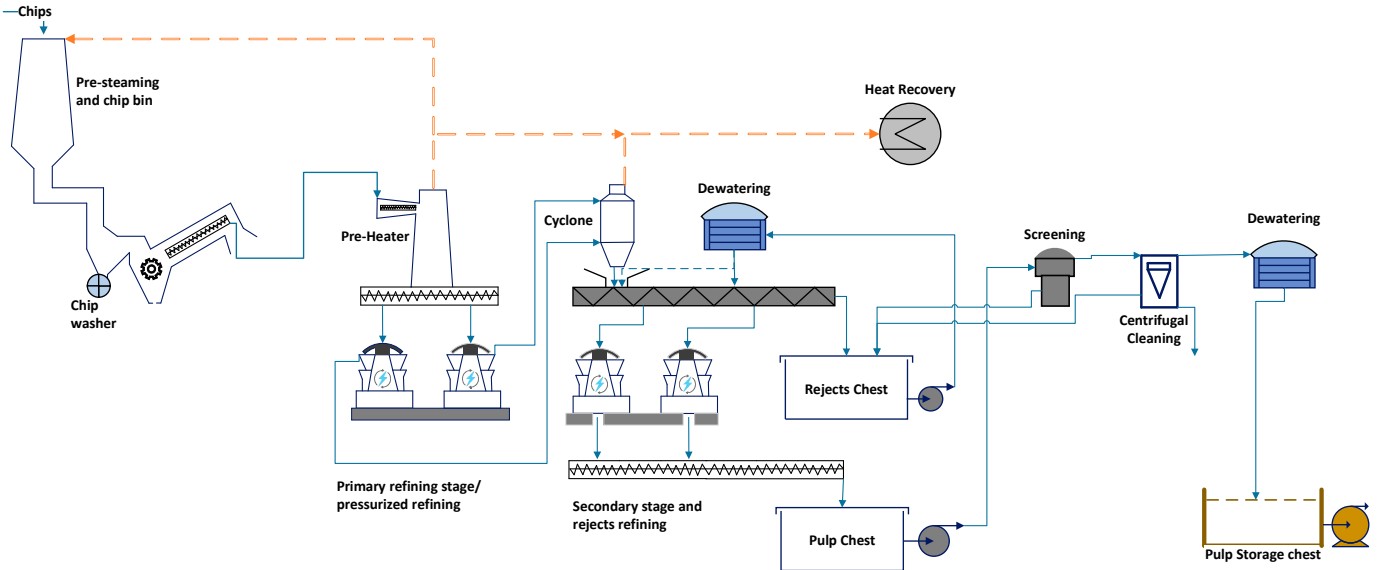

**Figure 1.** Sample layout of the thermo-mechanical pulp mill.

In general, the pulp and paper process is highly complicated and integrated [15]. The process consists of several interconnected operations and recycled loops. Moreover, there are many constraints such as fuel and power prices, environmental issues, availability, the performance of equipment, production schedules, and process variability that extremely harden the process of energy simulation and optimization. The refiner is the core and backbone of the thermo-mechanical pulping process. The refining process is stated as the most energy-consuming process at the TMP plant, accounting for close to 80% of the total energy consumption of the TMP process [16]. In these hydraulic machines, the rotor and stator crush and beat the pulp fibers in a repeated cycle. Although there are different configurations of the refiners in terms of technical details, relying on pulping process requirements, they have the same operational principle. There are several factors, such as segments operating time, rotational speed, crossing angle, refining gap, bar dimensions, and pulp consistency, influence the pulp properties and pulping energy consumption. Often, operating the refining process with worn plates/segments leads to a reduction in motor load as well as pulp quality. Nevertheless, the primary goal of the refining is always to keep the quality of pulp at the desired level, and a reduction in energy consumption should not affect the quality.

For the refining process's energy efficiency analysis, a simulation model is required to predict the essential refining variables for energy efficiency calculation. Due to the fact that a large part of the consumed energy in the refining process is converted into heat, the amount of recovered heat should also be considered as useful energy and should be reflected in the calculation of energy efficiency. Because of the large volume of refining heat production, the heat recovery unit is an integral part of the refining process, and in this article, the refining process is called a combination of refiner set and heat recovery unit. Therefore, the amount of produced steam in the refiner and recovered steam in the heat recovery unit are among the important variables in determining the refining energy efficiency. Since the amount of recovered heat depends on the heating demand of the heat recovery unit, heating demand is also one of the variables affecting energy efficiency, which in our case study is the heating demand from the district heating side. The volume of heating demand from the heat recovery unit can be a function of various factors. For example, if the generated heat is going to be used in the paper mill, the amount of recovered heat is a function of energy demand in the drying unit of the paper machine, or if the generated heat feeds the district heating system, the recovered heat is a function of the district heating demand or the site ambient temperature. Therefore, it is not always possible to recycle all the refining produced steam, and in some cases, when the heat demand is

low, the excess heat and steam are inevitably directed to the chimney (waste of energy). Therefore, in order to evaluate the energy efficiency of the refining process, the formation of an identification model that can estimate the amount of refining produced steam and electricity consumption in the grand electric motor according to the system operating condition and refining variables is of particular importance. The conventional energy efficiency analysis method is to calculate refining specific energy consumption, which is the ratio of refiner motor load to throughput. This is while, in this study, a new variable as corrected refining specific energy consumption (CRSEC) is introduced and practiced for better representation of the refining energy efficiency. In the calculation process of the CRSEC, the recovered energy from refining generated steam is also considered as useful energy and subtracted from refining motor load to calculate refiner energy input. In the studied case, generated heat is used for the district heating network. Refining motor load and refining generated steam are two essential variables for the corrected refining specific energy consumption (CRSEC) computation.

- Literature review

As stated earlier developing an accurate refining identification model is essential for energy efficiency analysis of the refining process. A mathematical simulation model, which summarizes a physical process into the mathematical equations, can be mechanistic, empirical, or a combination of both for a complicated process. A mechanistic model relies on physical principles and is applicable to a wide range of system operating conditions. The major problem with mechanistic models is that they often become complicated and time-consuming to develop for some processes. Empirical models are mostly based on fitting a mathematical equation to the experimental data. They are easier to produce but approximate the process over limited operating conditions. The other drawback of empirical models is that performing experimental tests is often impossible or expensive for some processes. Concerning empirical models, Talebjedi et al. [17] conducted a comprehensive study to simulate the refining process in the thermo-mechanical pulp mill using machine learning methods. Their results show that despite the high complexity of this process, which is affected by many variables, machine learning methods have great power in simulating the refining process. Their achievement confirms that combining the adaptive neuro-fuzzy inference system (ANFIS) model with the particle swarm optimization (PSO) algorithm shows the best performance in terms of accuracy compared to other heuristic optimization algorithms. Ciesielski and Olejnik's [18] findings confirm a high potential of artificial neural networks for the prediction of the quality and properties of the chemical pulps refined papers. Musavi and Qiao [19] recommended a radial basis function (RBF) neural network for pulp quality simulation and prediction in the digester. Their results prove an impressive prediction model efficiency in terms of accuracy of predicting pulp Kappa number. Regarding theoretical analysis, Ping et al. [20] investigated the computational fluid dynamic (CFD) simulation of a high consistency disc refiner with ANSYS CFX to reduce the energy consumption of the refining process. Their results show the potential of the CFD method in the analysis of the pressure and torque on the disc plate and calculating the power consumption of the refining process. Olson et al. [21] investigated the designing of a high-performance pulp screen rotor, benefiting from CFD simulation. They advanced and practiced a CFD simulation for designing an inventive multi-element foil rotor. Their results show that the new rotor provides 43% energy reduction over the state-of-the-art rotor technology. Khokhar [22] conducted computer simulation based on CFD modeling of a refiner with a simple groove. By considering pulp as single-phase Newtonian fluid with dynamic viscosity 100 times higher than water, Khokhar analyzed factors influencing the flow development in the rotor and stator of the refiner. The results show that the pressure difference is the main factor that affects flow in the rotor. Refiner speed has a significant effect on the flow in the rotor and a little impact on the flow in the stator.

- Novelty and paper contributions:

MATLAB Thermo-mechanical pulping simulation toolbox model is developed based on the combination of empirical and mechanistic methods. Apart from giving insights into

the process mechanism, an empirical-mechanistic model provides the process specifications, dynamics, and disturbances. The principal idea of developing this model was to generate a model to control the refiner motor load, pulp consistency, and motor vibration for the short-term time interval. Since this model is formulated for a short time window, variables affecting the refining process for the long-term analysis, such as plate erosion, are ignored. However, the optimal refining control strategy is highly dependent on the accuracy of the model introducing the physic of the process. Most TMP mills use a model predictive controller (MPC) to control the refining process. This controller requires an identification model to explain the refining process. The more accurate process simulation model leads to a more efficient MPC controller. Erosion of the refiner's discs is one of the most critical refining disturbance variables affecting the dynamic of the refining process. In this study, the longer-term effect of plate condition on refining behavior is studied by modifying the basic MATLAB Thermo-Mechanical Pulping Simulink toolbox for the use of longer-term analysis. Afterward, the developed model is employed to predict the refining motor load, generated steam, and *CRSEC*. The major development of the improved model is the ability to consider the effect of plate wear on the energy aspect of the refining process. Refining plate erosion affects the refining motor load, refining generated steam, and pulp quality. To our knowledge, there has not been a conducted research to evaluate the feasibility and accuracy of the MATLAB Thermo-Mechanical Pulping toolbox. Harinath et al. [23] used MATLAB Thermo-Mechanical Pulping toolbox to develop the refining optimal control strategy, so their focus was on developing the controller, not the model identification. Therefore, the novelty and paper contributions are summarized as follows:

- A new variable, corrected refining specific energy consumption (CRSEC), is introduced and practiced to better represent the refining process's energy efficiency.
- A MATLAB Simulink model is developed to refining energy simulation for the longer-term time interval.
- The effect of refining disturbance variables, mainly refining plate erosion, on refining motor load simulation is evaluated.

## 2. Materials and Methods

### 2.1. Refining Process

Refiners are mechanical devices correcting the morphology of fibers in wood. The source of energy for generating mechanical forces during the refining process is electricity. The high portion of the electric power consumption in refiners generates heat. Generated heat vaporizes the dilution water and also water mixed with the wood chips (moist contents), which turns the water-chip suspension into a fiber-steam suspension at the end of the process. As the composition of the material fed into the refiner varies during the refining process, the complexity of the problem increases [24]. A wood chip refiner configuration is either a single-disc or double-disc rotating plate (Figure 2). For the single-disc setting, there is one rotating disc and one stationary disc, and for the double-disc setup, two discs rotating in opposite directions. The typical rotation speed for a refiner disc induced by a grand motor is $1200 - 1800 \, rpm$. Recent refiners are available with up to 1.8 m diameters and 18,000 $Hp$ (13, 400 kW) of power supplied to each plate [25]. Higher refiner capacity requires a larger plate diameter and electric motor power. The plate gap is attentively controlled by a hydraulic cylinder to a distance of $0.05 - 2 \, mm$ [26].

In the refining process, wood chips are reduced to individual fibers. Refiner plates are designed to gradually break the wood chips and soften the lignin to separate fibers. The primary refiner is supplied by wood chips after wood chips pre-treatment (e.g., washing and steaming) utilizing a screw feeder. The purpose of the second stage refiner in the TMP process is to take care of the final development in the pulp qualities. High consistency refiners operate at a pulp consistency of 20–50%.

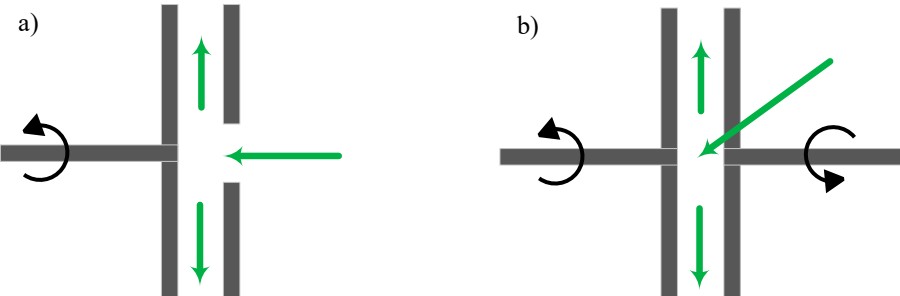

**Figure 2.** The common refiner types with pulp inlet and flow directions: (**a**) single disc, and (**b**) Double disc.

Case study: Figure 3 presents a description of the studied system. Moist wood chips are fed between the plate gap of the single-disc high consistency refiner. Dilution water is added to control pulp consistency. During the refining process, mechanical forces are transferred to the wood chips to soften the lignin and separate wood fibers. The studied refiner is the first stage refiner in a thermo-mechanical pulping mill with two parallel refining lines. Refiner nominal power is 7.5 MW, and the pressurized refining generated steam is being used to supply district heating. The generated heat in the refining process is directed to the heat recovery unit, and the excess heat is sent to the chimney. The recovered heat is used to supply the heat load of the district heating network. Therefore, its value is entirely dependent on the district heating demand or ambient temperature (since the district heating demand depends on the site ambient temperature).

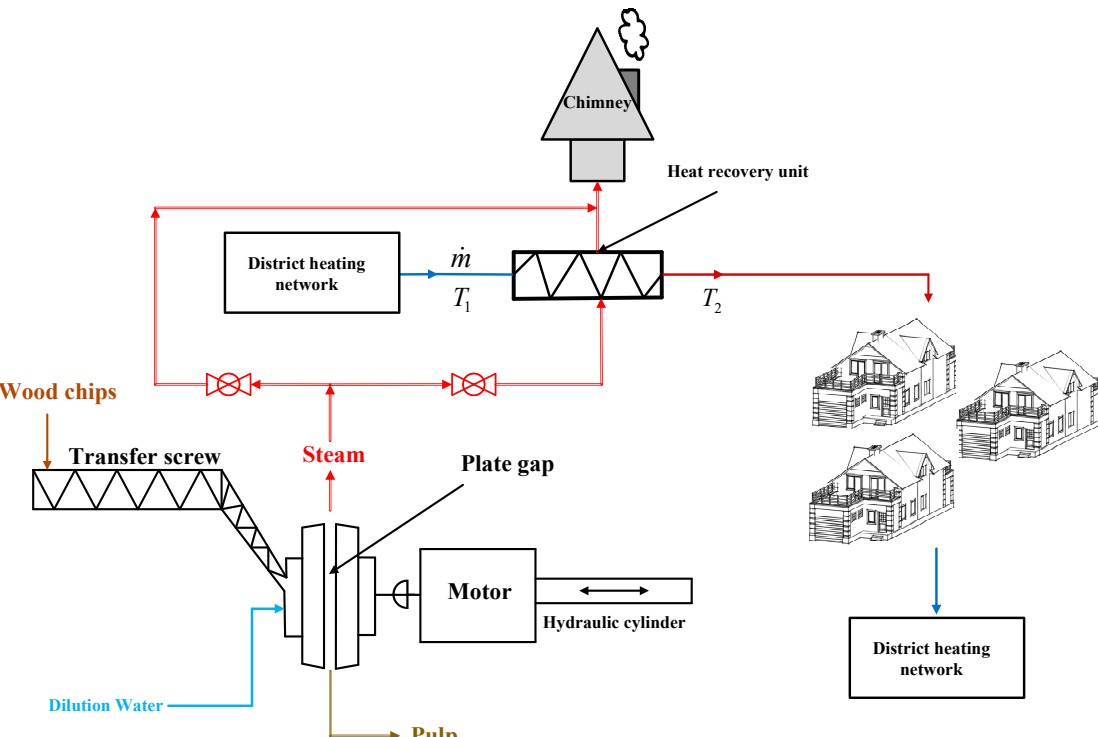

**Figure 3.** Scope and schematic of the studied system.

### 2.2. Principal Manipulated Variables

In previous researches, refiner plate gap, dilution water, and the number of wood fibers per unit volume entering the refiner (which is a function of chips transfer screw

speed and dilution water flow rate) are considered to be key variables for modeling the refiner motor load [23,27].

Chip transfer screw speed: The volumetric flow of the wood chips to the main refining line is controlled by the chip transfer screw feeder before the first stage refiner. The pulp mill uses primary feeder rotational speed to set the desired throughput. Changes in the screw feeder affect the dry fiber flows to the process.

Dilution water flow rate: Refining pulp consistency is a function of the dilution water flow rate. Pulp consistency in the refining zone is an essential variable for pulp quality estimation. Changes in the pulp consistency for a given refining specific energy consumption alters refining intensity, imposed specific energy, and finally, pulp quality. In terms of process stability, a high dilution water flow rate results in unstable refiner operation. Controlling the inlet refining dilution water flow rate is a straightforward way to keep the refining consistency on the desired value.

Plate gap: Refiner disc gap is set to the desired distance by either a hydraulic or electro-mechanical loading system. Plate gap measurement is an important operation action to control the refiner motor load and prevent plate clashing as well [28]. Plate gap variation affects the exerted mechanical forces to the wood material and specific energy consumption for a given production rate.

### 2.3. Disturbance Variables

Generally, disturbance variables are the type of variables that cannot be controlled and measured in many situations. Refiner plates operating conditions and inlet wood chips uniformity are the most dominant disturbances affecting the refining energy consumption and pulp quality.

Refiner plate wear: The nature of the refining process does not follow a stationary behavior because of the refiner plate erosion. Most often, refiner plates erode gradually based on the plate operation time and need to be renewed after hundreds of hours. However. in the case of plate clashing, plates cannot work anymore, and they should be changed immediately [29]. Generally speaking, new plates produce pulp with higher quality and freeness values for a given refining specific energy consumption. Short-term load variation rises for refiner with worn plates. Also, fully loading the refiner with eroded plates becomes increasingly hard and impossible in some cases. Preventing plates from eroding is impossible, but the erosion effect could be compensated by advanced manipulating the control variables to some extent.

Raw material quality: Chip bulk density, wood species, chip solid content are mentioned as the prominent disturbances for the raw material quality. Meanwhile, chip bulk density is figured as the main disturbance for the dry wood mass flow or production rate. Wood chips density variation straightly changes the pulp quality and specific energy consumption for a given motor load [30].

### 2.4. Principal Operating Variables

Basically, operating variables are the result of the influence of manipulated and disturbance variables. In the simulation model, they are considered as model outputs.

Refining motor load: An electrical motor is always integrated into the mechanical refiners to actuate the refiner discs. It is known that refining specific energy consumption has a great correlation to the pulp quality, and for a given production rate, the motor load is proportional to the specific energy consumption. Basically, motor load variation is a function of pulp consistency, plate gap, production rate, wood chips properties, etc.

Refining consistency: Refining consistency changes the ratio of the refining specific energy consumption and pulp quality. Refining dilution water flow rate is the most simple and dominant variable to control the refining consistency, although other variables such as motor load and production rate affect the consistency as well. For a given refining specific energy, variations in refining consistency lead to the different pulp quality [31].

Production rate: The oven-dry wood flow rate feeding to the refiner is called a production rate or throughput. The production rate is proportional to chip-transfer screw speed and could be easily computed by multiplying the chip-transfer screw speed by a constant value. Chip bulk density and chip solid content are the main disturbance variables affecting the production rate. Equation (1) calculates the production rate (ton/hour) by taking into account the wood chips properties variations [23].

$$P(R) = 0.06 k_p s_c d_c R \tag{1}$$

where $k_p (m^3/rev)$ is proportional constant, $s_c (\%)$ is chip solid content, $d_c (Kg/m^3)$ is chip bulk density, and $R(rpm)$ is chip-transfer screw speed, respectively.

Refining specific energy: Refining specific energy consumption $(RSEC)$ $(MWh/ton)$ is the refining energy consumption per produced ton of dry pulp. This value is the variant of energy efficiency and profoundly affects the pulp quality. Lower $(RSEC)$ indicates a more energy-efficient process for a given pulp quality. Mostly, $RSEC$ is a function of the four prominent refining variables, namely dilution water, feeder speed, plate gap, and refiner plate condition. Changes in the mentioned variables affect pulp quality, as well. The most prominent expression for $RSEC$ in literature is the ratio of the refining motor load to throughput. However, most often generated steam in the refining process is used for the drying in the papermaking process, and it is not waste heat. Using the refining generated heat for the district heating network is also a good action to increase refining energy efficiency. Therefore, a new variable as $CRSEC$ is introduced for better presenting the refining process energy efficiency than the conventional index $(RSEC)$, where the effect of refining recovered heat is reflected in refining energy efficiency calculation.

To summarize, $RSEC$ is a suitable value to present the energy efficiency of the refiner to produce pulp for a given quality. Also, higher $RSEC$ meaning higher pulp quality for a given production rate. However, $CRSEC$ is a good variable to represent the energy efficiency of the whole refining process, which consists of a refiner set and heat recovery unit (heat recovery unit is an inseparable part of the thermo-mechanical pulping mills). The energy input for the heat recovery unit is the energy coming from refining generated steam; therefore, lower heating demand from the heat recovery unit leads to excess heat, which must be diverted to the chimney. Increasing the flow of steam to the chimney reduces the energy efficiency of the process because the opportunity to recover heat is lost, and the generated heat is practically wasted. The following statement defines the corrected refining specific energy consumption $(CRSEC)$:

$$CRSEC = \frac{Input\ energy}{production\ rate} \tag{2}$$

where the expression for energy input is:

$$Input\ energy = ML - RH \tag{3}$$

where $ML$ is refining motor load, and $RH$ is the recovered heat in the heat recovery unit. Since recovered heat in the heat recovery unit depends on the district heating demand (in the case study) and refining generated heat, Equations (4) and (5) display the $RH$ calculation method:

$$RH = DHD \quad if \quad ERGH \geq DHD \tag{4}$$

$$RH = ERGH \quad if \quad ERGH \leq DHD \tag{5}$$

where $DHD$ is district heating demand, and $ERGH$ is the energy from refining generated heat. Figure 4 depicts the $RSEC$ of the studied refiner. The gradual decline of the $RSEC$ in the repetitive period represents the effect of plate erosion on refining motor load per unit of pulp production. From the graph, it is clear that higher plate operation time results in a higher level of the eroded plate and less refining motor load per unit of pulp production,

and lower pulp quality. Huge jumps in *RSEC* are due to changing the refiner plates. New plates produce pulp with higher quality and higher *RSEC* than worn plates.

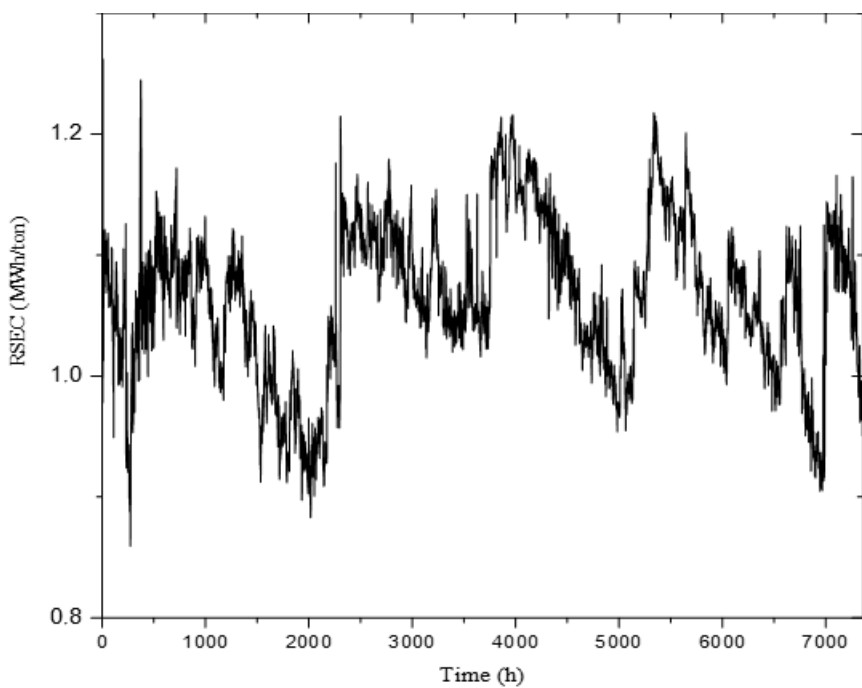

**Figure 4.** Measured RSEC.

### 2.5. High Consistency Refiner Energy Simulation

Basic MATLAB Simulink model: MATLAB Simulink model (presented in Figure 5) is a powerful tool to model a two-stage thermo-mechanical pulping refining process. Model inputs are feeder rotational speed, dilution water flow rate, and plate gap, while model outputs are refining motor load, generated steam, pulp consistency, and refiner vibration for each refining stage. Figure 6 gives a better description of the model for calculating first stage pulp consistency, motor load, and refining evaporated water. The main drawback of this model is ignoring the plate condition effect on refining behavior. Changing the refiner plate's condition creates disturbances that affect refining energy consumption and pulp quality. This might happen as a consequence of plate erosion. Plate operation time has a minor effect on refining behavior for short-term refining simulation after plate change. When refining plates are new and in good condition, the basic MATLAB Simulink model could be able to predict refining motor load with high accuracy. However, for long-term simulation, the basic MATLAB Simulink model works poorly in predicting refining characteristics because of ignoring the effect of refining plate condition on refining performance.

For this reason, the influence of plate segments operation time on refining behavior is modeled utilizing the real measured data from the thermo-mechanical pulp mill in a Nordic country. Pearson correlation coefficient reveals the high correlation between refiner plate operation time and refining motor load ($-0.6$). More time to use plates results in faster plate erosion and, consequently, lower motor load and pulp quality. The Pearson correlation indicates the extent of the linear relation of two variables. This number has the upper and lower bound of $+1$ and $-1$, respectively. A correlation of $-1$ means the linear descending link of data points in a scatter plot. In this case, variables are quite negatively linearly related. A correlation of 0 shows the zero degrees of the linear relationship between variables. Though, there might be a non-linear relation between the two variables. The correlation coefficient of 1 determines the perfect linear relation between two variables.

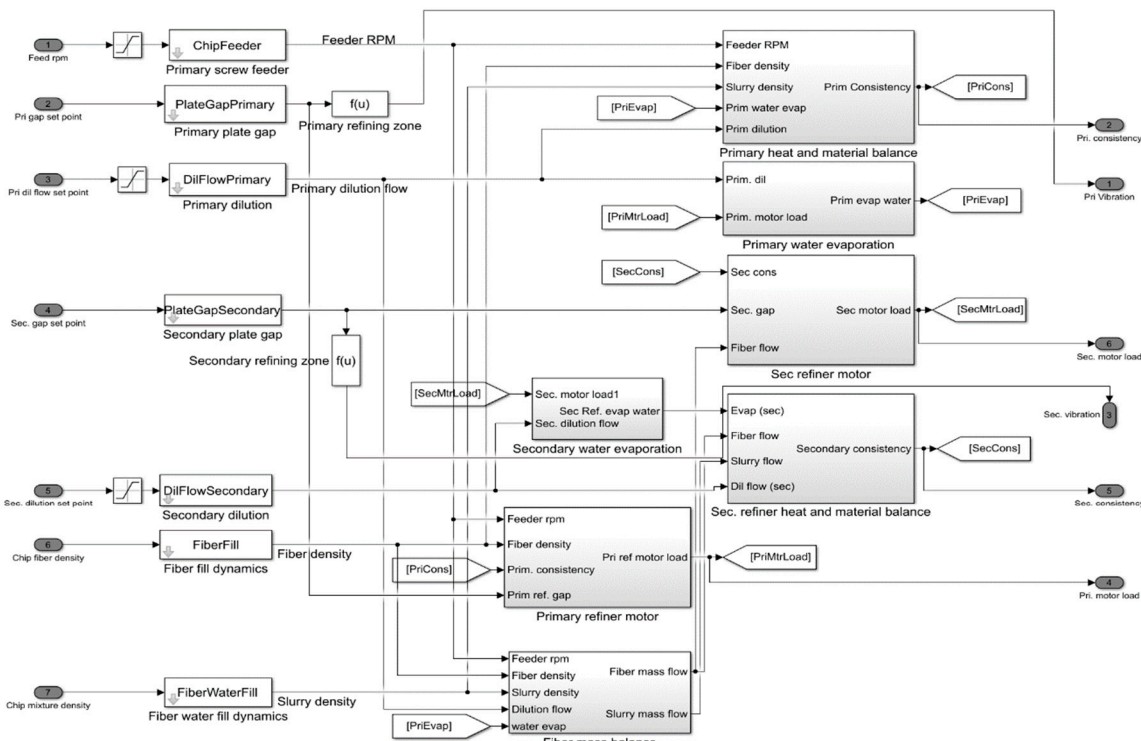

**Figure 5.** MATLAB Simulink toolbox model for the two-stage Thermo-mechanical pulping process (This model has been used in the case study of MPC Supervisory Control of a Two Stage Thermo-Mechanical Pulping Process, Model Predictive Control Toolbox, MATLAB R2019a).

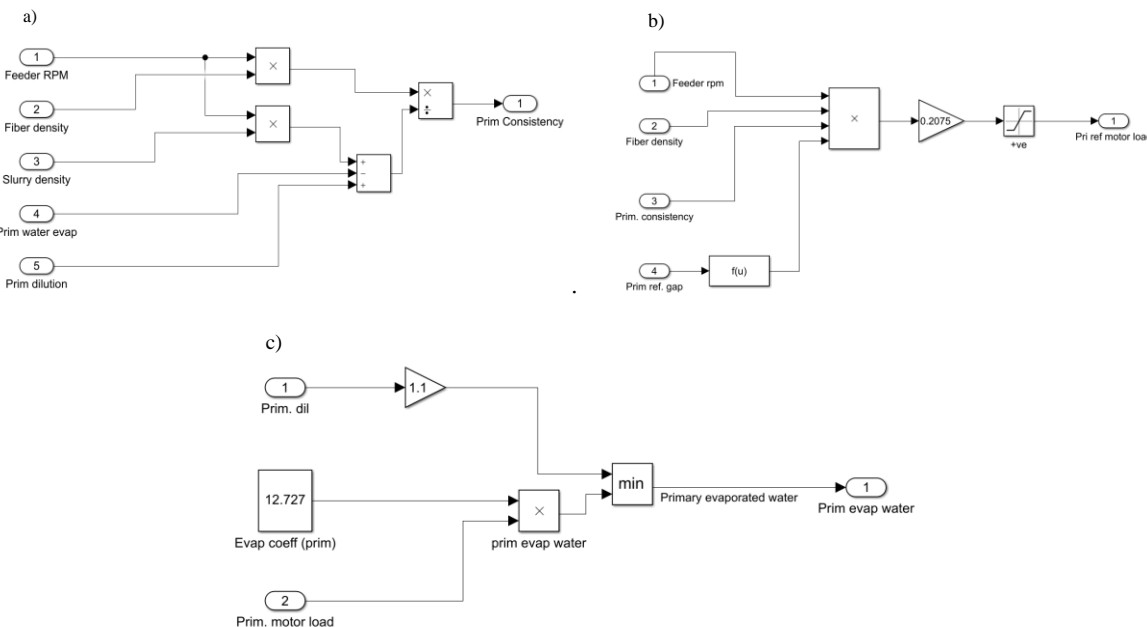

**Figure 6.** Relationship between refining variables and primary pulp consistency (**a**), primary motor load (**b**), and primary refining evaporated water (**c**). (MATLAB Thermo-mechanical pulping Simulink toolbox model).

Developed simulation model: Developed refining simulation model is a modified version of the basic MATLAB thermo-mechanical pulping Simulink toolbox model. The primary purpose of this model is to predict the refining energy variables for longer-term energy analysis. For a longer-term study, the refining plate condition plays an important role in refining behavior and should be taken into account. As stated earlier, the plate

condition is working as a strong disturbance for refining simulation. As plate operation time increases, plates become more eroded, and there would be more variation in refining motor load for a given production rate.

In order to model the refining plate erosion, a block diagram related to the refining motor load (in Figure 5) should be replaced by the new one, which introduces the motor load as a function of the plate operating time. Figure 7 is the modified version of the Simulink block diagram regarding the motor load prediction (Figure 6b). The new block diagram (deterioration function) has four coefficients to implement the effect of the refiner plate deterioration on the refining motor load. Since the amount of produced steam is calculated from the motor load, modifying the motor power also indirectly corrects the amount of produced steam. Figure 7 illustrates the new block diagram, where inside the mentioned black box is the plate deterioration function. The deterioration function coefficients are calculated by running an optimization model for the best motor load fit where the optimization error criterion is the mean least square error ($\varepsilon_1$). The rest of this chapter is devoted to explaining the optimization method. The refining motor load in the deterioration function (Figure 7) is corrected by Equation (6).

$$ML^{Corrected} = C_1 + C_2 \times ML^{Old} + C_3 \times POT + C_4 \times ML^{Old} \times POT \qquad (6)$$

where $C_i$ are model coefficients, $ML^{Old}$ is the predicted refining motor load by the Basic MATLAB Simulink model, $ML^{Corrected}$ is the corrected value of motor load by the deterioration function, and $POT$ is the refiner plate operation time. Model coefficients in Equation (6) are calculated by minimizing the model least square error, which is provided in Equation (7). It is possible to calculate the minimum of $\varepsilon_2$ instead of $\varepsilon_1$ since they behave in the same way for the minimization purpose. In order to obtain the minimum value of $\varepsilon_1$, derivatives with respect to the model coefficients must be set to zero as given in Equations (10)–(13). By rearranging Equations (10)–(13) in an implicit form, everything comes down to solving 4 by 4 linear system (AX = B) given in Equation (14), where there are four equations and four unknowns. It should be noted that since error (the difference between measured and predicted value) is convex, setting the derivatives to zero will find the minimum of the error function and not the maximum of it.

$$\varepsilon_1 = \sqrt{\frac{1}{n} \sum_{k=1}^{n} \left| ML_k^{Corrected} - ML_k^{Measured} \right|^2} \qquad (7)$$

$$\varepsilon_2 = \sum_{k=1}^{n} \left| ML_k^{Corrected} - ML_k^{Measured} \right|^2 \qquad (8)$$

where $ML^{Measured}$ is the measured value of refining motor load. Substituting Equation (6) in Equation (8) gives:

$$\varepsilon_2 = \sum_{k=1}^{n} \left| C_1 + C_2 \times ML_k^{old} + C_3 \times POT_k + C_4 \times ML_k^{old} \times POT_k - ML_k^{Measured} \right|^2 \qquad (9)$$

Equations (10)–(13) provide derivatives taken from $\varepsilon_2$ with respect to model coefficients ($C_i$) :

$$\frac{\partial \varepsilon_2}{C_1} = 0 \quad \rightarrow \quad \sum_{k=1}^{n} 2 \times \left| C_1 + C_2 \times ML_k^{Old} + C_3 \times POT_k + C_4 \times ML_k^{Old} \times POT_k - ML_k^{measured} \right| \times 1 = 0 \qquad (10)$$

$$\frac{\partial \varepsilon_2}{C_2} = 0 \quad \rightarrow \quad \sum_{k=1}^{n} 2 \times \left| C_1 + C_2 \times ML_k^{Old} + C_3 \times POT_k + C_4 \times ML_k^{Old} \times POT_k - ML_k^{measured} \right| \times ML_k^{Old} = 0 \qquad (11)$$

$$\frac{\partial \varepsilon_2}{C_3} = 0 \quad \rightarrow \quad \sum_{k=1}^{n} 2 \times \left| C_1 + C_2 \times ML_k^{Old} + C_3 \times POT_k + C_4 \times ML_k^{Old} \times POT_k - ML_k^{measured} \right| \times POT_k = 0 \qquad (12)$$

$$\frac{\partial \varepsilon_2}{C_4} = 0 \quad \rightarrow \quad \sum_{k=1}^{n} 2 \times \left| C_1 + C_2 \times ML_k^{Old} + C_3 \times POT_k + C_4 \times ML_k^{Old} \times POT_k - ML_k^{measured} \right| \times ML_k^{Old} \times POT_k = 0 \tag{13}$$

Writing Equations (10)–(13) in an implicit form gives:

$$\begin{pmatrix} n & \sum\limits_{k=1}^{n} Ml_k^{old} & \sum\limits_{k=1}^{n} POT_k & \sum\limits_{k=1}^{n} Ml_k^{old} \times POT_k \\ \sum\limits_{k=1}^{n} Ml_k^{old} & \sum\limits_{k=1}^{n} Ml_k^{old^2} & \sum\limits_{k=1}^{n} Ml_k^{old} \times POT_k & \sum\limits_{k=1}^{n} Ml_k^{old^2} \times POT_k \\ \sum\limits_{k=1}^{n} POT_k & \sum\limits_{k=1}^{n} Ml_k^{old} \times POT_k & \sum\limits_{k=1}^{n} POT_k^2 & \sum\limits_{k=1}^{n} Ml_k^{old} \times POT_k^2 \\ \sum\limits_{k=1}^{n} Ml_k^{old} \times POT_k & \sum\limits_{k=1}^{n} Ml_k^{old^2} \times POT_k & \sum\limits_{k=1}^{n} Ml_k^{old} \times POT_k^2 & \sum\limits_{k=1}^{n} Ml_k^{old^2} \times POT_k^2 \end{pmatrix} \times \begin{pmatrix} C_1 \\ C_2 \\ C_3 \\ C_4 \end{pmatrix} = \begin{pmatrix} \sum\limits_{k=1}^{n} Ml_k^{measured} \\ \sum\limits_{k=1}^{n} Ml_k^{old} \times Ml_k^{measured} \\ \sum\limits_{k=1}^{n} POT_k \times Ml_k^{measured} \\ \sum\limits_{k=1}^{n} POT_k \times Ml_k^{old} \times Ml_k^{measured} \end{pmatrix} \tag{14}$$

Therefore, model coefficients in Equation (6) can be calculated from Equation (15):

$$\begin{pmatrix} C_1 \\ C_2 \\ C_3 \\ C_4 \end{pmatrix} = \begin{pmatrix} n & \sum\limits_{k=1}^{n} Ml_k^{old} & \sum\limits_{k=1}^{n} POT_k & \sum\limits_{k=1}^{n} Ml_k^{old} \times POT_k \\ \sum\limits_{k=1}^{n} Ml_k^{old} & \sum\limits_{k=1}^{n} Ml_k^{old^2} & \sum\limits_{k=1}^{n} Ml_k^{old} \times POT_k & \sum\limits_{k=1}^{n} Ml_k^{old^2} \times POT_k \\ \sum\limits_{k=1}^{n} POT_k & \sum\limits_{k=1}^{n} Ml_k^{old} \times POT_k & \sum\limits_{k=1}^{n} POT_k^2 & \sum\limits_{k=1}^{n} Ml_k^{old} \times POT_k^2 \\ \sum\limits_{k=1}^{n} Ml_k^{old} \times POT_k & \sum\limits_{k=1}^{n} Ml_k^{old^2} \times POT_k & \sum\limits_{k=1}^{n} Ml_k^{old} \times POT_k^2 & \sum\limits_{k=1}^{n} Ml_k^{old^2} \times POT_k^2 \end{pmatrix}^{-1} \times \begin{pmatrix} \sum\limits_{k=1}^{n} Ml_k^{measured} \\ \sum\limits_{k=1}^{n} Ml_k^{old} \times Ml_k^{measured} \\ \sum\limits_{k=1}^{n} POT_k \times Ml_k^{measured} \\ \sum\limits_{k=1}^{n} POT_k \times Ml_k^{old} \times Ml_k^{measured} \end{pmatrix} \tag{15}$$

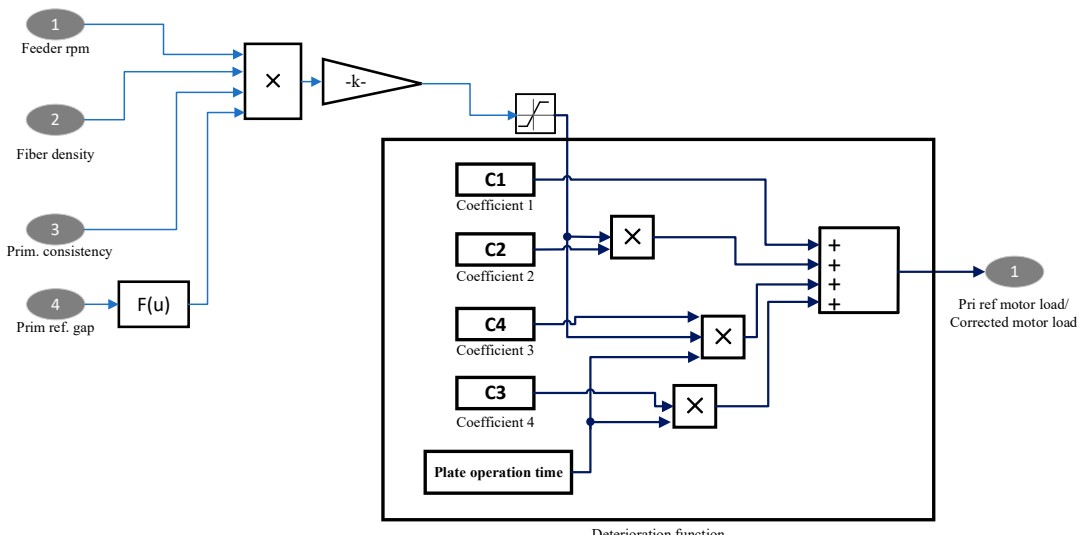

**Figure 7.** New motor load formulation.

## 3. Results and Discussion

For the model accuracy evaluation, the Coefficient of determination $(R^2)$ and Mean absolute percentage error $(MAPE)$ are used as two criteria to compare the simulation model to the actual measured value.

$$R^2 = \frac{\left[ \sum\limits_{k=1}^{n} (x_k - \overline{x})(y_k - \overline{y}) \right]^2}{\sum\limits_{k=1}^{n} (x_k - \overline{x})^2 \sum\limits_{k=1}^{n} (y_k - \overline{y})^2} \tag{16}$$

$$MAPE = \frac{1}{n} \sum_{k=1}^{n} \frac{|x_k - y_k|}{|x_k|} \tag{17}$$

where $x_k.y_k.\overline{x}.\overline{y}$ and $n$ are the measured data, simulation data, mean of measured data, mean of simulation data, and the number of data, respectively.

### 3.1. Refining Simulation Model

Basic MATLAB refining Simulink model: Plate operation time has a minor effect on refining behavior for short term refining simulation after plate change. Table 1 provides the statistical features of the basic MATLAB Simulink refining simulation model for short-term and longer-term analysis. From Table 1, it is clear that the basic MATLAB simulation model is performing well in predicting refining motor load (when plates are new and in good condition). Nevertheless, for longer-term simulation, the basic MATLAB Simulink model is not efficient in predicting refining characteristics because of ignoring the effect of refining plate condition. Therefore, for the longer-term simulation period, when changing the refiner plate and discs abrasion impose an effect on refining energy behavior, the basic MATLAB simulation model cannot predict the desired operational variable accurately.

**Table 1.** Basic MATLAB Simulink model results for short- and longer-term prediction of refining motor load.

| Parameter | Short-Term Analysis | Longer-Term Analysis |
|---|---|---|
| *MAPE* | 1.1% | 8% |
| $R^2$ | 80% | 30% |

Figure 8 shows the scatter plot of the basic MATLAB simulation model to predict the refining motor load for the short-term (24 h) and longer-term (200 h) horizon. Red lines are predicted variables obtained from the basic MATLAB simulation model, and black lines are values calculated based on the real measurement from the pulp mill. Figure 8b shows the lesser amount for the measured refining motor load than the estimated one because of the plate wear. This is due to the lack of consideration of the refiner plate erosion on the energy simulation of the refining process. Higher plate erosion leads to lower transferred mechanical energy to the pulp and reduces refining energy consumption. Therefore, in Figure 8b, red points, which are predicted values of the refining motor load (without considering the plate condition), are above the black spots (measured refining motor load).

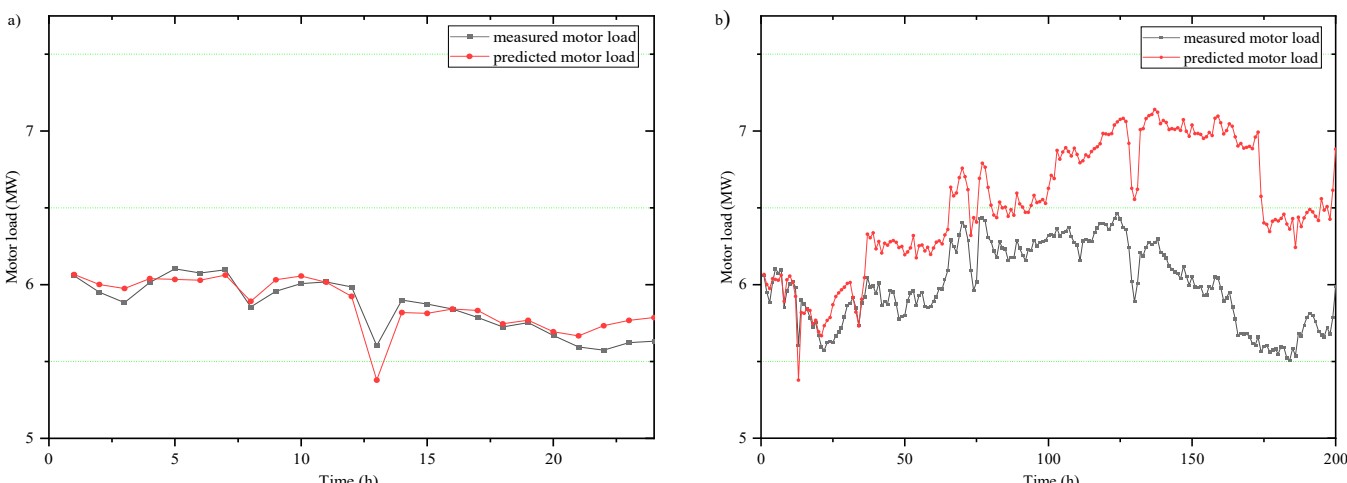

**Figure 8.** Measured and predicted (basic model) refining characteristics for: (**a**) short-term (**b**) long-term analysis.

Developed refining simulation model: Developed refining simulation model is a modified version of the basic MATLAB thermo-mechanical pulping toolbox. The main purpose of developing the basic model is predicting the refining energy behavior for longer-term analysis. For longer term analysis, the refining plate condition plays an important role and should be taken into account. As stated earlier, the plate condition is working as a strong disturbance variable for refining energy simulation. As plate operation time

increases, plates become more eroded, and there would be more variation in refining motor load for a given production rate. In this study, the simulation box related to the motor load calculation is corrected with three coefficients to model the effect of plate deterioration on refining motor load. Since refining evaporated water is being calculated based on the refining dilution water and refining motor load, correcting the refining motor load leads to the correction of the refining generated steam automatically. Therefore, the developed model can be used for steam prediction as well.

Four optimal simulation model coefficients are presented in Table 2. Due to the differences in the structure of the plate, the model coefficients fall within the given intervals, and no unique coefficient is obtained for all conditions. Even if the plates are of the same type, the slight variation in the quality of the plate manufacturing is an important factor for the differentiation of plates.

**Table 2.** Optimal coefficients for the developed refining simulation model.

| Coefficient | Min | Max |
|---|---|---|
| $C_1$ | $-1.48$ | $-0.65$ |
| $C_2$ | $0.92$ | $1.25$ |
| $C_3$ | $-0.00662$ | $-0.00232$ |
| $C_4$ | $0.012$ | $0.046$ |

Table 3 reports the statistical analysis results regarding the accuracy of the developed MATLAB Simulink model for estimating the refining motor load. The developed model reveals 160% and 78,75% improvement in simulation model determination coefficient and error, respectively. The modified simulation model can be used to predict the two essential parameters (refining generated steam and motor load) to compute *CRSEC* based on the input variables measured from the mill.

**Table 3.** Accuracy evaluation of the developed Simulink model.

| Parameter | Refining Motor Load (ML) | Difference% w.r.t Basic Model |
|---|---|---|
| *MAPE* | 1.7% | $-78.75\%$ |
| $R^2$ | 78% | 160% |

Figure 9 illustrates the developed MATLAB simulation model to predict the refining motor load for the longer period. Red lines are predicted variables obtained from the developed MATLAB simulation model, and black lines are the real measurement of refining motor load from the studied pulp mill. It is clear in the graph that the developed model has a higher accuracy in estimating refining variables, which shows the importance of the developed model to achieve the optimal refining control strategy.

*3.2. Energy Efficiency*

Pulp refiners generate a huge amount of heat, and most often, this heat is recovered to use for other purposes such as papermaking or district heating. For energy efficiency, the recovered heat should be considered as useful energy. For this reason, a new variable as corrected refining specific energy consumption (*CRSEC*) is introduced and practiced for better representation of the refining energy efficiency. In the calculation process of the *CRSEC*, the use of recovering energy from refining generated steam is considered as useful energy and subtracted from refining motor load. Therefore, *RSEC* is a suitable value to present the energy efficiency of the refiner to produce pulp for a given quality. Also, higher *RSEC* means more top pulp quality for a given production rate. However, *CRSEC* is a good variable to represent the energy efficiency of the whole refining process, which consists of a refiner and heat recovery unit (heat recovery unit is an inseparable part of the thermo-mechanical pulping mills).

District heating demand: The heat sink for the studied refiner is district heating. Using generated heat for the district heating network advances to higher refining energy efficiency or lower *CRSEC*. The district heating demand differs during the year based on the ambient temperature and other factors, but the ambient temperature is the most influential variable. Figure 10 represents the district heating demand according to the time (for 7360 data).

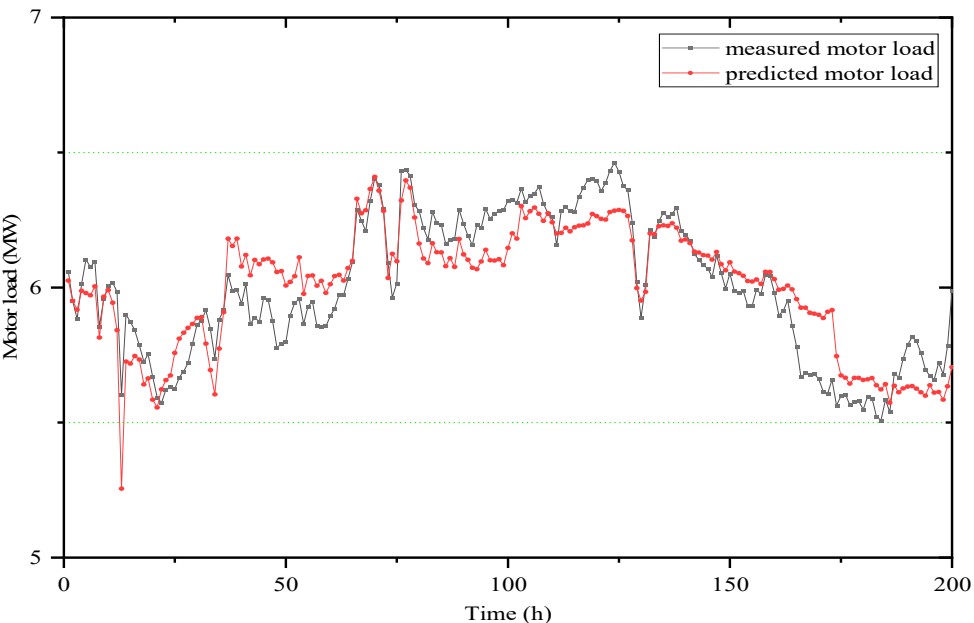

**Figure 9.** Measured and predicted (developed model) refining characteristics for long-term analysis (plate 1).

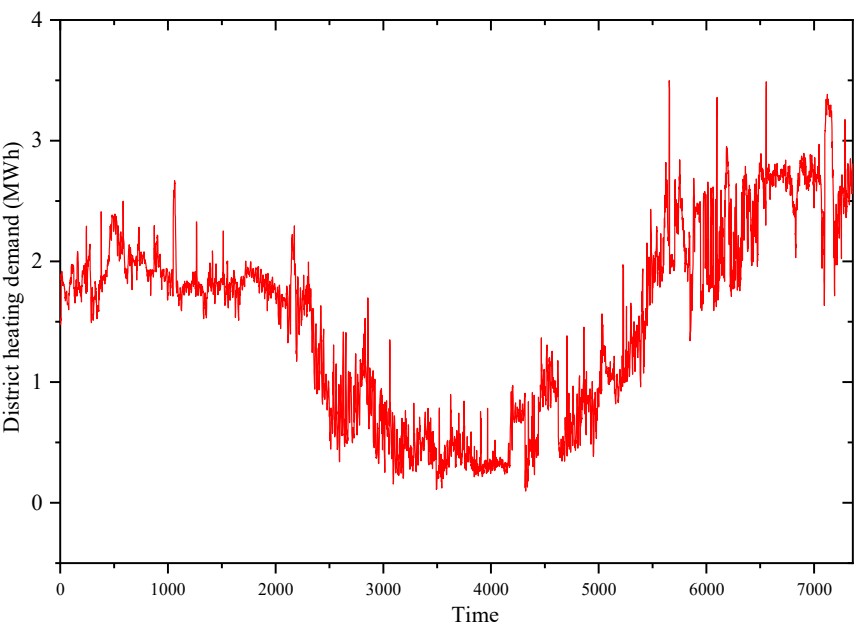

**Figure 10.** Annual district heating demand.

Corrected refining specific energy consumption (CRSEC): As mentioned earlier, a large amount of steam is generated during the refining process, which can be consumed as useful energy by heat energy recovering in the heat recovery unit. Therefore, to calculate the energy efficiency, the effect of the amount of recovered energy from the generated heat must also be considered and reflected. This effect is applied in this research by introducing

a new variable as corrected refining specific energy consumption. Figure 11 shows the obtained values for corrected refining specific energy consumption (CRSEC), and Figure 12 illustrates the differences between two variants of energy efficiency, which are *CRSEC* and *RSEC* for the almost whole year. In other words, Figure 12 shows the percentage of energy efficiency improvement when the heat recovery unit is used to supply the district heating network. Therefore, Figure 12 shows the percentage of energy efficiency change while using the proposed method of calculating energy efficiency instead of the conventional technique of measuring energy efficiency. Comparing Figures 10 and 11, it is clear that the energy efficiency of the refining process has a great correlation with the demand from the heat recovery unit, which is district heating demand in our case study. This relationship works oppositely, so that the higher the heat demand, the higher the energy efficiency of the process due to the increase in recovered energy. As a consequence, higher district heating demand, which is equal to the less ambient temperature, leads to the refining process's higher energy efficiency. According to Figure 11, the refining process energy efficiency in winter is higher than in summer, so that the average CRSEC is about 0.75 in winter, 1.02 in summer, and 0.8557 for the whole year. Less CRSEC means more energy efficiency for a certain amount of pulp quality. Therefore, about a 27% improvement in energy efficiency occurs in winter than summer, which is related to an increased amount of recovered heat.

Annual average *RSEC* and *CRSEC* are obtained as 1.0934 and 0.8557 (MWh/ton), respectively. This reduction is equal to a 22% improvement in energy efficiency by using refining generated steam to supply the district heating network. In other words, *RSEC* is a suitable value to present the pulp quality (most often higher *RSEC* meaning higher pulp quality) for a given production rate or energy efficiency of refiner for a given pulp quality, whereas *CRSEC* is a good value to represent the energy efficiency of the whole refining process, which consists of refiner set and heat recovery unit for a given pulp quality. As shown in Figure 12, due to higher heating demand in winter, the amount of steam directed to the chimney is reduced, which results in less waste energy, and increases the refining energy efficiency.

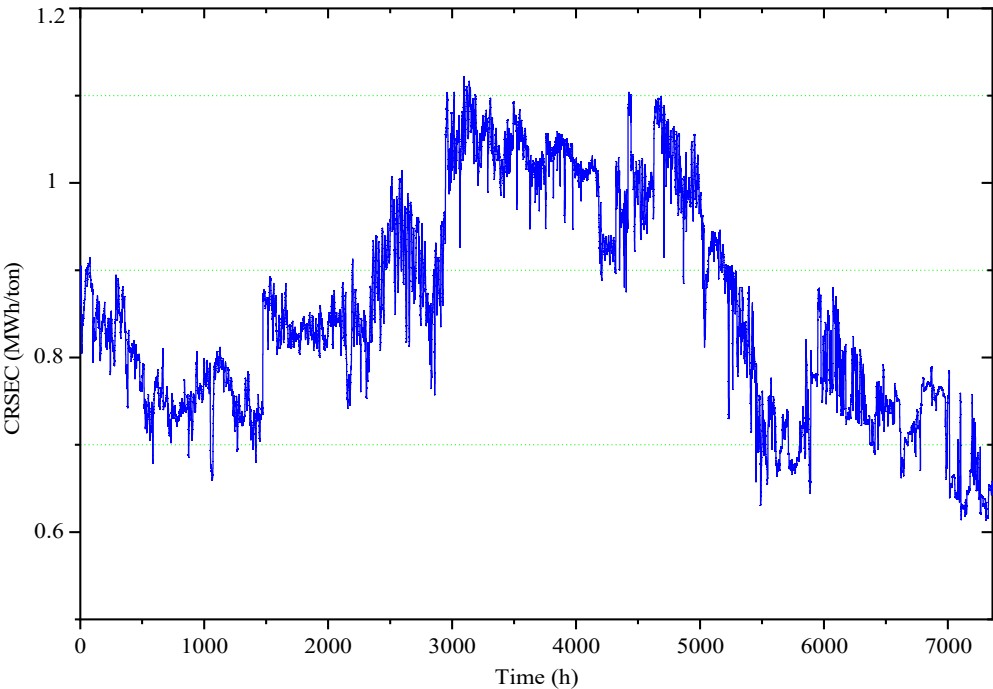

**Figure 11.** Corrected refining specific energy consumption (CRSEC).

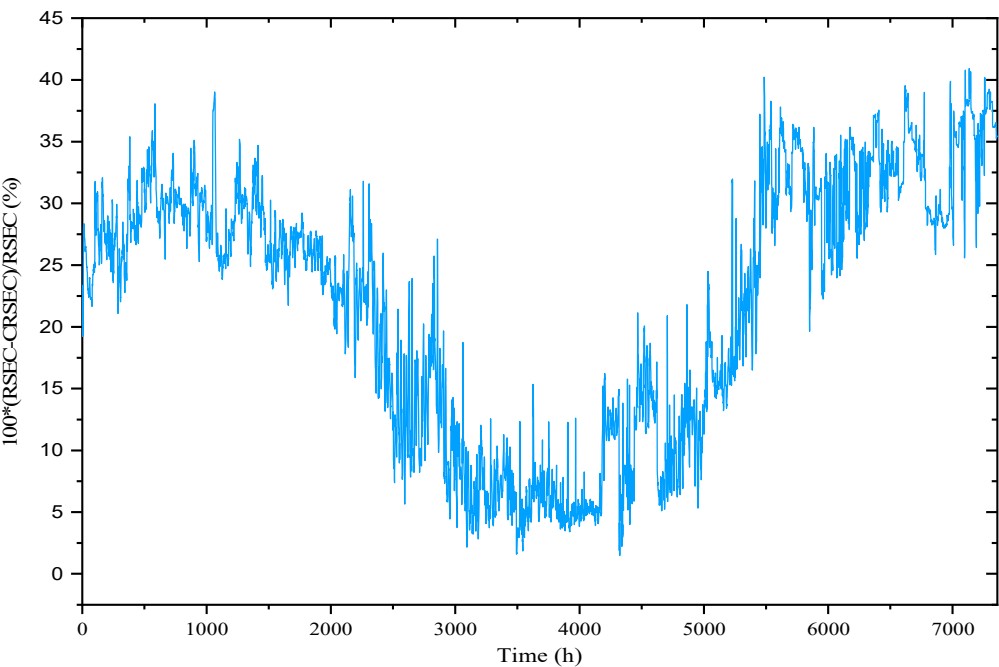

**Figure 12.** Comparing different methods for energy efficiency calculation.

## 4. Conclusions

Refining is the most energy-intensive process in thermo-mechanical pulping. Refiners consume a huge amount of electricity to convert wood chips into pulp. Refiners also produce steam, mostly used for drying units in the papermaking process (in the studied case, the generated heat supplies district heating). Since the heat recovery unit is an inseparable part of the refining process, for energy efficiency evaluation, refining generated steam should be considered as useful energy. The introduced CRSEC variable is a variant of the energy efficiency of the whole refining process, which consists of a refiner set and heat recovery units. To calculate the refining energy efficiency, a simulation model is developed to estimate the refining motor load and generated steam for calculating the CRSEC based on refining variables. The refiner plate condition is an important refining variable affecting the refining motor load and generated steam. This important refining variable has been disregarded in basic MATLAB refining simulation toolbox for refining energy identification model. The basic MATLAB model only works for short-term time intervals when plate condition has no effect on refining behavior. For this reason, the plate deterioration function is added to the model for simulating the plate erosion. In this research, the refining plate erosion is modeled based on the operation time of using the same plate for the refining process by modifying the basic MATLAB simulation model. The developed model is a modified version of the basic MATLAB thermo-mechanical pulping toolbox, where the effect of plate deterioration on refining behavior is considered. The developed model gives the correlation coefficient of 78% and MAPE of 1.7% for estimating the refining motor load for 200 hours' time period.

The difference between RSEC and CRSEC shows the potential for energy efficiency improvement while using refining generated heat for district heating. Results show an annual 22% improvement of the refining energy efficiency by the use of a heat recovery unit to supply district heating compared to the absence of the heat recovery unit. In other words, in the studied mill, the energy efficiency of the whole first stage refining process (including refiner and heat recovery units) is 22% higher than the energy efficiency of the refiner alone. District heating demand correlates with ambient temperature. Due to higher district heating demand in the winter, there is more potential for heat recovery, which leads to refining energy efficiency improvement. This improvement is about 27% in our case study. As future works, the generated model can be used to evaluate how the refining

identification model accuracy could affect developing the optimal refining control strategy, the energy-saving opportunity from better refining control, and a sustainable refining process. This model is not able to predict and simulate the quality of the paper based on refining variables. Since quality and energy are highly correlated in thermo-mechanical pulping, this issue can be considered a drawback of the model, and in future research, the model could be upgraded to estimate the paper quality parameters as well as the energy.

**Author Contributions:** Conceptualization, methodology, software, writing—original draft, formal analysis, writing—review & editing, B.T.; data curation, funding acquisition, project administration, supervision, writing—review & editing, T.L.; conceptualization, investigation, funding acquisition, supervision, writing—review & editing, H.H.; funding acquisition, writing—review & editing, E.V.; funding acquisition, supervision, project administration, writing—review & editing, S.S. All authors have read and agreed to the published version of the manuscript.

**Funding:** The authors gratefully acknowledge the support of the Academy of Finland, grant number 315020.

**Institutional Review Board Statement:** Not applicable.

**Informed Consent Statement:** Not applicable.

**Data Availability Statement:** Restrictions apply to the availability of these data.

**Acknowledgments:** We acknowledge having the opportunity to present our research at the 15th SDEWES Conference 2020 Cologne, Germany [32].

**Conflicts of Interest:** The authors declare no conflict of interest.

## Abbreviations

| | |
|---|---|
| *MAPE* | Mean-absolute percentage error |
| *CFD* | computational fluid dynamic |
| Variables | |
| *R* | Correlation coefficient |
| $R^2$ | Determination coefficient |
| $C_i$ | ith model coefficient |
| *Ml* | Refining motor load |
| $Ml^{old}$ | predicted refining motor load by the Basic MATLAB Simulink model |
| $Ml^{corrected}$ | corrected value of motor load by the deterioration function |
| $Ml^{measured}$ | measured value of refining motor load |
| *POT* | plate operation time |
| *RH* | Refining recovered heat in the heat recovery unit |
| *ERGH* | Energy from refining generated heat |
| *RSEC* | Refining specific energy consumption |
| *CRSEC* | Corrected refining specific energy consumption |
| *DHD* | district heating demand |
| Indices | |
| *i* | model coefficients index |
| *k* | data point index |
| *n* | number of data |

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
