# Peer review of "Energy Efficiency Analysis of the Refining Unit in Thermo-Mechanical Pulp Mill"

_energies, doi:10.3390/en14061664_

Round 1

Reviewer 1 Report

This paper argues that recovered heat from a pulp mill can be recycled to provide heat for surrounding areas, which increases the energy efficiency of the mill. Some comments about the paper:

  • Break the introduction into two sections, an introduction and a literature review. The introduction is too long and it take the reader too long to find out the contributions of the paper. The bullet points at the end of the section are a nice way to summarize the contributions, though.
  • The authors could do a better job explaining what a pulp mill is. I got the sense that is primary purpose was making paper and cardboard, but I also get the sense that it made other things as well. For instance, in Figure 3, is T2 part of the usual pulp mill configuration?
  • The innovation of this paper is treating refined generated steam as recoverable energy, which improves the average energy efficiency of a pulp mill by 22%. Why don't pulp mills do this?  Does it require any additional inputs that might offset this gain? For instance, the authors talk about needing a heat recovery unit. What it the cost of running this? Does running the heat recovery unit do anything to the quality of the paper? The authors mention this in the conclusion but leave it to future research. My main question is, why don't previous simulation models consider recovered steam energy and why do pulp mills not do so either?

Author Response

This paper argues that recovered heat from a pulp mill can be recycled to provide heat for surrounding areas, which increases the energy efficiency of the mill. Some comments about the paper:

1- Break the introduction into two sections, an introduction, and a literature review. The introduction is too long, and it take the reader too long to find out the contributions of the paper. The bullet points at the end of the section are a nice way to summarize the contributions, though.

Answer:

Thank you for your comment. You are right, and the required changes have been made to the text. So, two subsections under the title of Literature review and Novelty and paper contributions are added to the introduction.

2- The authors could do a better job explaining what a pulp mill is. I got the sense that is primary purpose was making paper and cardboard, but I also get the sense that it made other things as well. For instance, in Figure 3, is T2 part of the usual pulp mill configuration?

The innovation of this paper is treating refined generated steam as recoverable energy, which improves the average energy efficiency of a pulp mill by 22%. Why don't pulp mills do this?  Does it require any additional inputs that might offset this gain? For instance, the authors talk about needing a heat recovery unit. What is the cost of running this? Does running the heat recovery unit do anything to the quality of the paper? The authors mention this in the conclusion but leave it to future research. My main question is, why don't previous simulation models consider recovered steam energy, and why do pulp mills not do so either?

Answer:

Actually, the main idea of this paper is to develop a model that can calculate the amount of the refining generated steam based on the manipulated variables and reveal the correlation between energy efficiency and refining variables such as refining plate gap, dilution water, production rate, amount of generated steam and hourly demand for the heat in the heat recovery unit. These heat recovery units are already existing in all mills to supply paper machine demand or district heating demand. But the fact that how the refining variables could affect the refining energy efficiency is not clear for the mills' operator. Therefore, this paper suggests a refining identification model that can simulate the refining process from the energy perspective where the effect of some refining disturbance variables (such as plate erosion) are considered in the model for higher refining simulation accuracy. Since the steam generation measurement is hard and, in most mills, it is not measured, this model can help have a good estimation of hourly refining generated steam that can be used in the heat recovery unit.

Reviewer 2 Report

Would it be possible to make exergy analysis for the system studied?

If yes, which software and/or software modification should be required?

Author Response

Would it be possible to make an exergy analysis for the system studied?

If yes, which software and/or software modification should be required?

Answer:

You made a good point. Yes, it would be nice to do the exergy analysis in the thermomechanical pulp mill. But because of the complexity of the process, this analysis is not an easy task. At the moment, we do not have a strong idea of what is best for exergy analysis, but there are different commercial software, such as ASPEN Plus, or coding the thermodynamic laws and equations in MATLAB.

Reviewer 3 Report

The paper is quite well structured, the research process is clearly described, but some further improvements are required, as it follows:
- The abstract should state provide a concise description of the working framework and results, it is too long. 
- The relevance of the results would be good discussed broader, adding to a comparison with other studies carried out previously in the scientific literature should also be added. 
- the names and manufacturers of the equipment and software used should be added.

Author Response

The paper is quite well structured, the research process is clearly described, but some further improvements are required, as it follows:

1- The abstract should state provide a concise description of the working framework and results, it is too long.

Answer:

Thank you for the comments. Although the abstract is 244 words which is compatible with the Journal criteria for the Abstract length (250 words max), we did shorten the Abstract slightly to 200 words.

- The relevance of the results would be good discussed broader, adding to a comparison with other studies carried out previously in the scientific literature should also be added.

Answer:

Thank you for the comment. We have done our best to discuss the results in the most understanding way. In our opinion, further explanation of the results causes repetition, which is not desirable for the reader. In the introduction, there is a subsection of the literature review where the previous studies have been addressed.

- the names and manufacturers of the equipment and software used should be added.

Answer:

Thank you for the comment. We have mentioned in the text that all our analysis has been done by MATLAB software. Unfortunately, we are not permitted to publish process-related data and information because of our agreement with the industry.